# Designing for Green and Grey: Insights from Single-Use Plastic Water Bottles

**DOI:** 10.3390/ijerph19031423

**Published:** 2022-01-27

**Authors:** Taesun Kim, Sang-Don Lee

**Affiliations:** 1Department of Industrial Design, ERICA Campus, Hanyang University, Ansan 15588, Korea; 2Department of Environmental Sciences & Engineering, Ewha Womans University, Seoul 03760, Korea; lsd@ewha.ac.kr

**Keywords:** single-use plastic water bottles, environmentally friendly, socially inclusive, older adults

## Abstract

Recognizing modern society’s multiple risks, this study examines single-use plastic water bottles at the intersection of environmental degradation and societal carelessness for the elderly. While prioritising economic profits and plastic waste, we have neglected bottles’ typical poor openability for older people. Thus, we evaluated the openability of bottles with environmentally friendly and socially inclusive designs in South Korea by comparing older and younger adults’ experiences. Integrating different attributes than existing studies that analyse opening torque or one-handed opening, the test results show that older adults experience the poorest two-handed openability when bottles have both a weight thickness lower than 14.42 g and an easily squeezable bottle structure. In South Korea, companies advocate eco-friendliness by valuing lighter weight with less plastic and support user-friendliness by adopting easily squeezed sidewall patterns, while the industry maintains broad opening torque regulations; however, we show this combination exceeds older users’ capabilities. That is, for openability, bottles need to keep a weight thickness greater than 12 g, abandon easily squeezed sidewall patterns or reduce the opening torque regulation range to 100 N-cm. These socially favourable but ecologically adverse measures will be sustainable when the efficient linear economy transitions to an effective circular economy.

## 1. Introduction

Just as the COVID-19 pandemic entails a series of threats, such as health crises, economic declines, and a large amount of disposable plastic waste, the manufacturing risks in modern society are complex problems related to a multidimensional system of advanced industrialisation [1,2,3,4]. Such risks, with a wide range of conflicts and uncertainties, require an approach encompassing the economic, environmental, and societal dimensions of sustainability. Nonetheless, many studies deal with a single dimension of sustainability because each aspect is complicated in itself, and the natural environment and human society appear to be distinct matters. Thus, there are few studies concerning both ecological (green) and social (grey) issues together [5,6], except cases focusing on the benefits from the environmental volunteering of the elderly [7,8].

However, in South Korea (hereafter Korea), there is not enough time to resolve issues one at a time, as the green and grey problems the country faces are severe and simultaneous. Korea was already the world’s largest plastic-consuming country per capita before the pandemic. For polyethylene terephthalate (PET) alone, the country’s consumption was 4.9 billion bottles of approximately 300,000 tons in 2017 [9]. With the coronavirus outbreak, Korea’s plastic waste in the first half of 2020 averaged 850 metric tons a day, up 16% from 732 tons a year earlier [10]. In the meantime, the country is also suffering a dramatic change in the demographic landscape in the form of a rapidly ageing population. In 2020, its population aged over 65 accounted for 16.4% of the domestic population [11], and without active interventions, this proportion will increase to 40.1% by 2060 [12]. This expansion of the elderly population leads to an increase in the number of older people living alone or only with their spouse [13], which implies that there is a need to improve the surroundings for their independent living and thus empowerment in their lives.

Hence, we look at the everyday conditions entangled with environmental degradation and the design exclusion of older people. To make our surroundings ecological and inclusive, this study examines single-use plastic water bottles and asks the following questions:What are more decisive bottle attributes of openability to increase inclusivity for the elderly in the ordinary situation of a two-handed opening, not in artificial conditions of a one-handed opening?What measures related to designing the bottles can help us solve a complex problem in the competition between the ecological and social sustainability perspectives?

### 1.1. Greying South Korea’s Enabling Environment for the Elderly

The global population included 382 million people over 60 years old in 1980 and 962 million people in 2017 [14], and this figure is expected to be approximately 2.03 billion people in 2050 [15]. In terms of the population proportion, older adults represented one out of nine in 2018, but this figure will become one of five in 2050 [16]. Population ageing occurs in countries at various levels of economic development, and thus, it is an inevitable trend in every country.

In particular, Korea is ageing at one of the fastest rates worldwide [17]. In 1970, its population aged 65 years or older accounted for only 3.1% of the entire domestic population; this figure roughly doubled to 7.2% in 2000, and it doubled again to 14.2% in 2017 [18]. Based on this rate, the country is expected to become the second oldest worldwide by 2050 [19]. In addition, the percentage of Korean senior citizens living alone has risen as well. Between 1960 and 2010, this proportion increased from 1% to 10% among men and 3% to 31% among women [20]. These changes imply that the immediate environment of everyday life must be addressed to promptly raise seniors’ inclusivity for independent lives, different from the past. 

Since 2013, the Korean government has responded to the demand with age-friendly initiatives offering age-friendly physical outdoor infrastructures, such as low-floor buses, renovation of bus and subway stations, and LED-installed crosswalks for older adults, focusing on major cities [21]. This measure is in line with the WHO’s Age-Friendly Cities and Communities initiative launched in 2006 to promote active ageing and life satisfaction through elderly-friendly public facilities and services; however, the fiscal constraints of the public sector discourage more robust actions. For example, the Age-Friendly Initiative of Seoul incurred an expense increase in its annual budget from 3.7% in 2012 to nearly 7% in 2015 [22]. This increasing economic burden on the central and local governments indicates that the age-friendly movement needs to enter its second phase to spur changes and find a breakthrough. 

Given the limited ability to build an age-friendly environment due to the public sector’s financial stress, private sector engagement for stabilisation is an option. According to the Korea Chamber of Commerce and Industry (KCCI), only 11% of Korean companies have played in the silver economy, and 65% have no plan to enter the market [23]. To establish the second stage of the age-friendly movement, we need to erase economic scepticism. In Korea, the silver economy producing and providing products designed for senior citizens amounted to approximately USD 62.9 billion in 2020, jumping from USD 23.7 billion in 2012, with a compound annual growth rate (CAGR) of 13.0% [24]. 

Given the concern of the sustainability of the initiative, partnership with the private sector will create spill-over effects of expendability alongside behavioural or policy changes. When older people can build their active participation through supportive settings for their independent lives, living longer can be recognised as worthy of celebration, and “the benefits of the longevity dividend” can be maximised [16] (p. 159).

### 1.2. Single-Use Plastic Water Bottles: The Intersection of Ecological Perspectives and Social Values

The increase in bottled water consumption is fuelled by, along with lifestyle changes due to urbanisation, a pervasive trend towards health, safety, and wellness [25,26], such that some women and older people have been found to drink only bottled water [27]. The global bottled water market grows by 10% each year and shows the fastest growth in Asia and South America [28]. The market of Korea, one of the Asian countries, has also shown steady growth since its formation in 1995 [29]. Its size was approximately USD 400 million in 2015, then increased by 15.5% from the previous year to USD 600 million in 2016 [30], and became USD 677 million in 2018 [31].

While the market expansion of bottled water highlights the plastic waste crisis, there is another issue of bottles linked to ecological stringency. The problem is the poor openability of the bottles, especially for people with weak hand strength. Even though older people and even some young and non-disabled adults struggle to open the bottles, our society has neglected their difficulties. Industries have avoided addressing the issue by offering various jar openers, from small rubber discs to expensive gadgets. Nonetheless, there are no perfect tools to satisfy people who need help with grip, two strong hands, leverage strength, or all three at once. Each tool or device can help only some people by accommodating a particular type of difficulty. The countless tools themselves serve as evidence of the magnitude of this problem.

Academia has paid attention to the increasing openability issue [32,33]. Including initial studies identifying the overall relation between torque exertion and cover diameters [34,35,36], most studies have focused on specifying the amount of force that a person can apply under a particular artificial condition, in which subjects twist off the lids fixed to a torque metre device and with only one hand (see Figure 1). Without considering cap height, for 29 mm diameter caps of NyQuil medicine bottles, men and women who are 62 years old or older can apply, on average, 2.04 Nm and 1.05 Nm, respectively, and for 27 mm diameter caps of Coca-Cola bottles, the mean torques are 1.51 Nm and 0.92 Nm [37]. For caps 30 mm in diameter and 21 mm in height and 40 mm in diameter and 17 mm in height, the starting torques are 1.54 ± 0.19 Nm, and 0.58 ± 0.10 Nm, respectively [38]. These results indicate that the force people can apply to the lids increases with the lid diameter. Wrist-twisting torque in elderly individuals increased by 0.061 Nm per mm increase in lid diameter in the 31–74 mm range, but the strength decreases with jar lids with a diameter equal to or greater than 74 mm [39]. For most people, the optimal lid diameter is 73 mm, and the force-*generating* capacity is 1 Nm [40].

Studies disclose different attributes influencing torque exertion to open bottles and jars. People can exert greater strength with squares than circular lids between 20 mm and 50 mm diameter and with greater heights [41]. Deeper grooves in the lids causing increased static friction between the hand and the bottle closure is a factor that lets the elderly perceive gripping to be much easier [38], but they do not assist in opening torque. There were no significant effects of lid roughness at diameters of 31, 55, and 74 mm [39,42] or 48–86 mm [43]. This implies that the cap ridge (knurl) appearing on commercially available lids is an inadequate design feature for increasing people’s torque output.

While many studies examine the conditions to open covers, few have investigated the conditions with two hands. A two-handed opening in which one hand holds the bottle’s body and the other grips the bottle top allows most people to use higher strength. More specifically, the maximum power of a two-handed opening is 2 Nm with a 66 mm diameter lid [44]. A study of two-handed opening using equipment shows that the correct use of eight different jar openers to open jars is often still inferior to using bare hands alone for younger subjects, and aged people over 65 experience the same problems as younger people [45].

Despite abundant research on openability, the bare two-handed opening of 28 mm diameter screw caps remains unaddressed. Considering two-handed opening as the typical way to hold and open bottles and the most common lid diameter in the global and Korean bottled water markets, studies must be carried out to determine more influential attributes supporting openability in the general setting to reflect the reality of everyday life.

## 2. Materials and Methods

### 2.1. Materials

To understand the routine difficulties seniors encounter, single-use plastic water bottles were investigated. As shown in Table 1, the bottles tested in this study had 28 mm diameter screw caps and could hold 500 mL of water, representing popular and standardised bottle features in South Korea. This study considered the market share of bottled water products and their design features to effectively select water bottles representing the domestic market of South Korea. The market consisted of approximately 30 brands of domestic or imported products; however, significant brands comprised approximately 60% of the market (i.e., Samdasu 39.8%, Icis 12.3%, and Backsansu 8.5%) [31].

A water bottle is a simple product dependent on only some attributes: bottle shape, bottle thickness by weight, bottle patterns (helpful to squeeze or not), cap height, and cap ridge (or grooves). Considering a mutually exclusive and collectively exhaustive selection scheme and the market share in South Korea, five bottled water products were selected, and each bottle represented a particular type of design. Additionally, opening torques were measured by bottle type. Each type of mean torque was calculated based on 30 bottle measurements by bottle type. Opening torques were measured by a digital torque metre. Table 2 details each bottle’s characteristics.

### 2.2. Questionnaire Development

This study adopted the following process to develop a questionnaire to identify differences in expected and experienced difficulties in the design of water bottles and caps. First, relevant questions were collected from the literature on the perceived difficulties and user experience (UX) of daily things [46,47]. Second, the preliminary questions were reviewed and revised by four UX design professionals, referring to a task scenario to open the bottles, which allowed opportunities to verify the content validity of the questions. Third, a pilot test using the reviewed questions was carried out with five people to find any potential errors in wording, terminology, and conciseness. Reflecting the findings from the three steps stated above, the final version of the questionnaire was developed.

The questionnaire consisted of three parts: subjects’ demographic information, the experience before opening each bottle, and the experience after opening each bottle. The demographic questions included age, gender, and hand strength related to grasping objects. In terms of expected experience, the before-opening questions asked participants how easy each bottle was held and lifted. The after-opening questions focused on how preferable the participants considered each bottle’s attributes and their overall satisfaction with the bottle. Each question, except those regarding demographic information, was rated on a five-point Likert scale ranging from 1 (strongly disagree) to 5 (strongly agree). Table 2 presents the structure of the developed questionnaire.

### 2.3. Test Administration

To collect data on the experience of opening the water bottles, tests were administered at subjects’ community houses or workplaces to help them feel comfortable. Before answering the questions, each subject was given the task scenario and a task outline to ensure that they understood the study purpose. Each subject was asked to open the five different water bottles, with the brand names hidden and the labels removed. As the test proceeded, the three subtasks of grasping, lifting, and opening a bottle were explained by test helpers and performed by each subject. Immediately after each trial, subjects were asked to describe their experience in the questionnaire developed for the study. This study obtained subjects’ written informed consent to participate in the test, following the ethics and safety requirements of the institutional review board. To manage unexpected problems during the test, test helpers were present at all times. The subjects were offered an incentive to participate in the test, a gift voucher worth KRW 10,000, equivalent to USD 10. This study used SPSS version 26 (IBM Corp., New York, NY, USA) for the statistical analysis of the test data.

### 2.4. Characteristics of the Test Subjects

The tests were conducted to determine expected and experienced ease-of-use in opening the water bottles, with 72 subjects gathered through convenience and snowball sampling. They were divided into two groups: younger healthy adults and older healthy adults. The initial subjects of the younger group were contacted through the author’s network and then asked to recruit others through their networks. As a result, unintentionally, there was none of the age group of 51–60. There were no additional efforts to gather the age group members considering the absence of this age group would not cause any significant effects in terms of muscle loss with ageing. Muscle mass begins to decrease around the age of 50, but it accelerates after 60. The rate of physiological changes from ageing in the elderly over 65 is more than 25%, and over 80 is about 50% [48]. The other group comprised 38 older people aged more than 60 years who could independently perform daily tasks. This elderly group was recruited through the help of a manager at a community house in the Daegu–Gyeongbuk area in South Korea. Of the subjects, 58.33% were female and 41.67% male, reflecting the gender ratio imbalance of the elderly population in South Korea (see Table 3). 

## 3. Results

### 3.1. Experience Difference in Opening Plastic Water Bottles by Age Groups and Bottle Types

Differences in experienced ease-of-use in terms of lifting, holding, and opening single-use plastic water bottles were determined based on two variables: age group and bottle type (based on bottle attributes). Two-way analysis of variance (two-way ANOVA) was conducted with Scheffé post hoc comparisons at 0.05. With a large sample (>30), the test results showed that experienced ease of use was affected by age and water bottle features. Table 4 and Table 5 show the results based on the variables.

We found statistically significant differences in experienced ease of use by age (f = 14.802, *p* < 0.001) and bottle type (f = 14.506, *p* < 0.00), but the interaction between these predictors was not significant. The Scheffé post hoc test revealed substantial pairwise differences between Bottle A (M = 3.69) and Bottle C (M = 2.90), Bottle A (M = 3.69) and Bottle E (M = 3.19), Bottle B (M = 3.96) and Bottle C (M = 2.90), Bottle B (M = 3.96) and Bottle D (M = 3.35), and Bottle B (M = 3.96) and Bottle E (M = 3.19).

The mean scores of experienced ease of use by age and bottle attributes are plotted in the line graph in Figure 2. The plot shows that regardless of age group and level of hand function, users consider certain bottle types preferable when completing the task of opening the bottle caps. While the higher scores for Bottles A and B indicate that they have some positive attributes, the lowest scores for Bottle C imply negative characteristics. Despite having different bottle shapes (Bottle A: rounded square, Bottle B: circle), both bottles have a thicker bottle wall with a regular cap height and cap ridges. In contrast to the others, Bottle C has a thinner bottle wall with a short cap and an easy folding structure pattern. Bottle E, similar to Bottle C in some aspects, is a lighter bottle with an easy folding design and was ranked fourth in terms of ease of use among the tested bottles. Bottles C and E received statistically lower scores than Bottles A and B, and they made up the lowest score group.

### 3.2. Influential Factors of Opening Plastic Water Bottles

A regression analysis was performed to identify factors affecting the overall ease-of-opening experience. It showed that different factors influenced the experience of the two groups. For the younger adults, the important predictor was hand strength (*β* = 0.58, *t* = 8.13, *p* = 0.00), while for the older adults, significant predictors were bottle thickness (*β* = 0.21, *t* = 2.76, *p* = 0.01) and hand strength (*β* = 0.22, *t* = 2.81, *p* = 0.01). Table 6 provides the detailed results of the regression analysis.

## 4. Discussion

### 4.1. Water Bottle Features as Inclusive Enablers for the Elderly

Through ANOVA, we can empirically identify different degrees of ease of opening bottles by age and bottle type, and with regression analysis, we can confirm the difference in influential factors of the bottle opening experience by age group. The ease score gap between ages suggests an age-specific factor, and the similarity between the two score patterns implies a common factor affecting both age groups. Hence, it is necessary to compare and discern bottle attributes to raise the design inclusivity of the bottles.

First, a comparison with previous studies on bottle features and openability under the one-handed opening condition is presented below. In line with the result that the ridges of lids smaller than 31 mm for better grip do not influence torque exertion [39,42,43], the unique cap ridges of Bottle C have no substantial impact and cannot completely compensate for any negative effects of other attributes in the two-handed settings, such that Bottle C has the lowest score. In contrast to most studies neglecting cap height, a case reporting the cap height correlation with torque levels [41] requires clarification of the effectiveness of the variable. As if mirroring the uncertain value of cap height, the shorter caps of Bottle C and Bottle E show their influence through lower ease scores, but across all bottles tested in this study, cap height turns out to be a false predictor for the overall experience of openability.

Under two-handed opening in the ordinary setting, this study discovers the significance of bottle thickness by weight as a new driver affecting the experience of openability, especially for the elderly. In the general context of the two-handed bottle opening, one hand grips the bottle cap, and the other hand grasps the bottle sidewall (see Figure 3). If we cannot exert enough twisting force on the cap, we hold more tightly to the sidewall. Thus, older people who cannot apply enough force to the bottle cap naturally hold the sidewall more tightly, which explains why they favour a thicker and more solid bottle wall that resists the hands’ holding and twisting strength. Moreover, a quickly squeezable structure design does not change bottle wall thickness but supports the inductive effect of the thickness, which is observed in this study. Bottles C, D, and E, with lower scores than Bottles A and B, have lighter weights than the 14.42 g criterion of the global initiative for reducing plastic waste and guaranteeing safe storage, distribution, and use for 500 mL PET bottles [49]. Among these three bottles, Bottles C and E belong to the lowest ease group, and they have squeezable bottle patterns, unlike Bottle D.

However, the fundamental reason people feel difficulty and seize the sidewall too tightly is that the opening torque exceeds their hand strength, which leads us to think about whether the regulation of opening torque applied to the bottles is proper. While Korean regulation specifies the range of bottle opening torque from 0.70 to 1.80 Nm [50], the Japan Food Packaging Association requires approximately 1 Nm for plastic bottles [51]. Although the information on the twisting force of aged people for 28 mm bottle caps is scarce, in a European study with 200 subjects (mean age: men 72.8 years old, women 74.1 years old), their mean torques exerted to 27 mm diameter caps were 1.51 Nm and 0.92 Nm, respectively [41], and an average 70-year-old adult has strength similar to that of a ten-year-old child [52]. In a recent study, Japanese subjects approximately 60 years old (mean age: 61.5) showed approximately 1.05 Nm torque when opening bottle caps with 28 mm diameter [53].

Given older adults’ physical capacities and barriers and other countries’ opening torque regulations, we can suggest a feasible measure: if ecological circumstances allow, the industry needs to make bottles not exceeding the thickness criterion of 14.42 g, or if society wants to keep bottles lighter than 14.42 g, the industry needs to abandon the easy-to-squeeze structure. Furthermore, the government needs to revisit the opening torque range, narrowing it down to approximately 1 Nm. These measures encompassing actions of the private and public sectors are relevant to the complexity of risk in advanced societies and become an intersection of efforts to build an inclusive environment. Although many older adults live alone and purchase single-serve items for convenience [54], over 70% of 2000 people in the retirement age group were found to abandon a product they cannot open, and 91% asked for assistance to open a package [55]. Our assumption that a daily product is available at any supermarket and thus is easily replaced with a different one is a foolish approach to design exclusivity, as it prevents commodities from becoming enablers for the elderly.

### 4.2. Business-Compatible Strategies with Ecological–Social Sustainability through the Circular Economy

In designing single-use plastic water bottles, some of the socially inclusive measures for the elderly suggested in this study are expected to ignite debate about ecological damage; adopting heavier, thicker bottles to enhance openability can lead to more plastic consumption and plastic waste. In the face of the imminent climate crisis and the unexpected threats that come with it, immediate actions to slow global warming are imperative. We cannot put aside this grave ecological mission under any circumstances, nor can we take a piecemeal approach focusing only on environmental degradation without considering social sustainability dimensions that bring design exclusion. Designing ‘environmentally friendly and socially inclusive’ bottles is at the intersection between the environmental and social sustainability dimensions.

In this context, companies have to create business strategies that are compatible with ecological–social sustainability. To date, corporations have concentrated on improving the efficiency of doing more by using less, which explains why the bottled water companies of Korea put efforts solely towards bottle weight reduction, leading to the poor openability for the elderly; however, even if the decrease in plastic consumption continues successfully, if natural resources continue to be used over time, they will be depleted one day. This finiteness of nature requires us to abandon the efficiency focus of a linear economy to the effectivity focus of a circular economy [56], in which not only the ecological impact is minimised but also the economic, ecological, and social impacts are positive [57]. The circular economy to increase effectivity pursues a system change emphasizing three elements at the core: closed cycles of recycling raw materials, components and products, renewable energy, and systems thinking [58].

Likewise, although a bottle thick enough to resist against the holding pressure of the hand is unfavourable to the environment, this fault can be conquered by adopting the circularity model: increasing the recycling rate of the PET water bottles, using biodegradable or recycled plastic bottles, and implementing a life cycle management system of PET bottles. Although the separate collection rate of PET bottles in Korea amounted to 80% in 2017, the recycling rate was only approximately 45% out of the whole production amount of 28.6 thousand tons. This low recycling rate was blamed for barriers against obtaining pure material; bottles with any food and liquid residue inside, adhesive labels, or chemicals accounted for 35% of unrecyclable bottles, and coloured bottles accounted for 20% [59]. Hence, to increase the recycling rate, the Korean industry can introduce new bottle designs, such as label-free bottles, non-adhesive label bottles, or completely transparent bottles. In 2021, beverage companies, including Nongshim Co. (food and beverage company, Seoul, South Korea), Dongwon F&B Co., Seoul, South Korea, Lotte Chilsung Beverage Co., Seoul, South Korea, Sansu Beverage Co., Seoul, South Korea, Jeju Province Development Corp., Seoul, South Korea, Coca-Cola Korea, Pulmuone Saemmul Co., Seoul, South Korea, and Hite Jinro Co., Seoul, South Korea, signed an agreement with the Ministry of Environment of Korea to expand the recycling of packaging materials, and companies with a combined share of 74% of the nation’s mineral water market planned to release label-free products, engraving their brand names on plastic bottles in the year [60]. The ministry stated that if all 4.2 billion plastic water bottled products sold in 2019 had been replaced with label-free products, it would have saved 24.6 million tons of plastic waste [61]. Another way to support recycling is producing bottles made of biodegradable or recycled plastic. Lotte Chilsung released recycled PET (rPET) bottles [62], and Evian, a foreign actor in the Korean bottled mineral water product market, introduced a prototype bottle made entirely of rPET [63]. Furthermore, the Korean bottled water industry has decided to construct product-as-a-service systems to effectively renew their bottles. For example, in cooperation with SK chemicals (SK), the Jeju Province Development Corp (JPDC) of Samdasoo has launched the recycling ecosystem of manufacturing, collecting, and recycling bottles [60]. In the system dubbed ‘Green Whole Process,’ JPDC and SK each have responsibilities. JPDC collects its transparent plastic bottles from all over the country through a home delivery service platform and provides bottles to SK. Then, SK puts the bottles into rPET and produces copolyester, a raw material for cosmetic containers, textiles, and home appliances [64]. With the recycling system, companies expand the recycling value chain by revitalizing the Korean circular economy.

In the dilemma of single-use plastic water bottles where two different perspectives compete, the criteria for a successful design should not be limited to the profitability of the manufacturers [65]. A successful design should encompass the two unlike domains of green and grey and have an ecological contribution and sociocultural significance at the same time. Hence, instead of continuing to produce thinner bottles resulting in plastic consumption reduction but yielding non-openability for the elderly, we need to transition to an integrated product-service combination loop in which the bottles are endlessly rebirthed into new products.

## 5. Conclusions

A chain of sustainability-threatening crises places humanity in a quandary between environmental peril and societal hazards linked to population ageing. To confront a routine challenge in daily life, this study investigated a solution to the friction between the sustainability of the green and the grey economies by comparing the experience of opening single-use plastic water bottles between two different age groups. The study findings corroborate bottled water product inadequacy as an enabler of older adults’ independence of living and determined specific bottle attributes to enhance bottle openability. Moreover, this study sheds light on the transition from the efficiency-oriented, selective approach of the linear economy to the effectiveness-focused, integrated circular economy to compensate for the negative effects of the preferred bottle option on the environment and support the elderly.

The industry and design community should outline plans that support the coexistence of human beings and nature on Earth through an explorative perspective and the elimination of barriers, such as industry inertia [66,67]. As product developers and designers, we are empowered to bridge gaps between conventional policies, ecological imperatives, and challenges associated with population ageing and create a better future for people surrounded by various products. While this statement is true, it may also sound boring; however, have we, as product designers, considered the boring truth?

## Figures and Tables

**Figure 1 ijerph-19-01423-f001:**
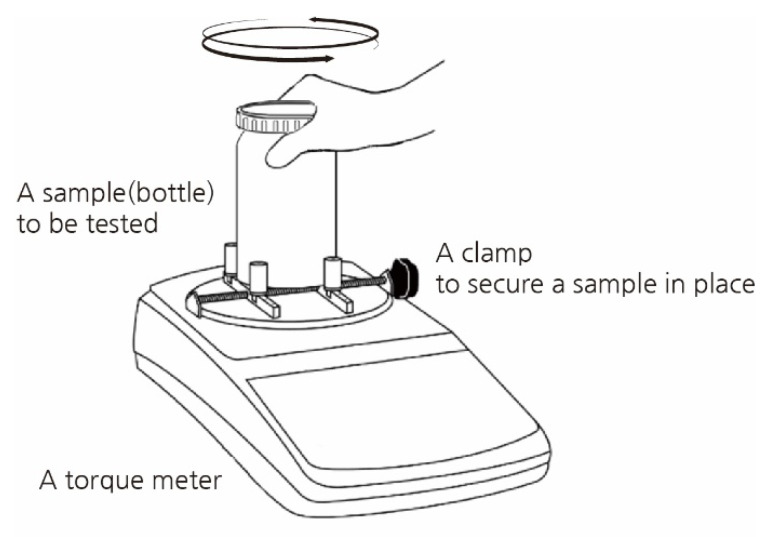
A setting for one-handed opening test with a torque metre in existing studies.

**Figure 2 ijerph-19-01423-f002:**
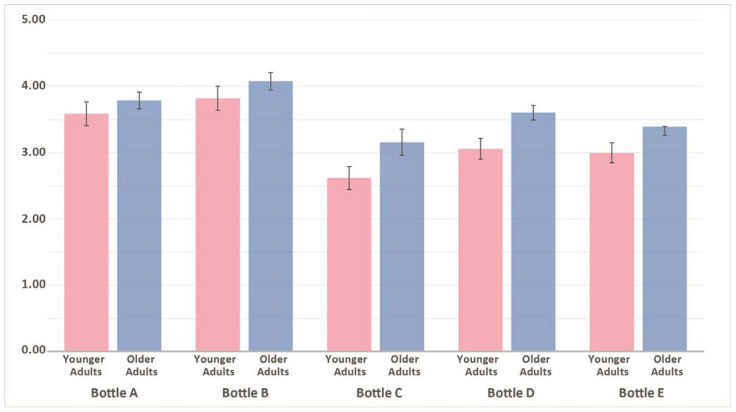
Comparison of the experience opening water bottles between age groups.

**Figure 3 ijerph-19-01423-f003:**
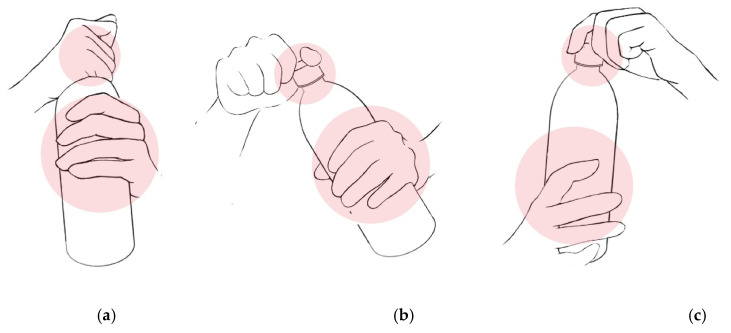
Two-hand positions to open bottled water products in ordinary settings: (a) Holding the top of the bottle and twisting with the cylindrical pinch); (b) holding the centre of the bottle and twisting with the lateral grip; (**c**) holding the top of the bottle and twisting with the chuck grip.

**Table 1 ijerph-19-01423-t001:** Questionnaire structure to measure the ease of use of single-use plastic water bottles.

Question Category	Question Topic
Demographic information	Age
	Gender
	Hand strength
Before-opening experience	Visual appearance
	Visual preference
After-opening experience	Bottle thickness
	Cap height
	Cap ridges
	Perceived difficulties to open
	Feeling of spilling bottle content
	Overall preference

**Table 2 ijerph-19-01423-t002:** Product features of the single-use plastic water bottles (500 mL) tested in this study.

Item	Bottle A	Bottle B	Bottle C	Bottle D	Bottle E
Image	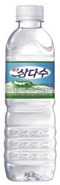	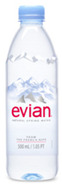	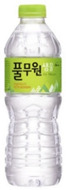	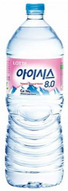	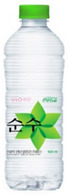
Overall Bottle shape	Rounded square	Circle	Circle	Circle	Circle
Bottle thickness by weight	18 g	20 g	12 g	13 g	14 g
Easy-to-squeeze pattern	None	None	Yes	None	Yes
Cap height (upper + lower part)	17.1 mm (13 + 4.1)	19.7 mm (12.7 + 7)	13.2 mm (9.7 + 3.5)	17.2 mm (11.7 + 5.5)	16.3 mm (12.3 + 4)
Cap Ridges	Regular	Regular	Wide	Regular	Regular
Mean opening torque (min/max)	107.04 (95.84/119.04)	117.01 (106.40/127.68)	146.83 (126.40/160.48)	129.92 (122.40/135.20)	122.29 (119.68/126.24)

**Table 3 ijerph-19-01423-t003:** Demographic characteristics of subjects.

Classification		Younger Adults	Older Adults	Total, *n* (%)
Age (years)	≤30	14	-	14 (19.44)
	31–40	14	-	14 (19.44)
	41–50	6	-	6 (8.33)
	51–60	0	-	0 (0)
	61–70	-	15	15 (20.83)
	71–80	-	16	16 (22.22)
	≥81	-	7	7 (9.72)
	Total	34	38	72 (100)
Gender	Male	16	14	30 (41.67)
	Female	18	24	42 (58.33)
	Total	34	38	72 (100)
Hand strength (kg)	≤10	0	4	4 (5.56)
	11–20	0	8	8 (11.11)
	21–30	17	13	30 (41.67)
	31–40	7	12	19 (26.39)
	41–60	9	1	10 (13.89)
	≥61	1	0	1 (1.39)
	Total	34	38	72 (100)

**Table 4 ijerph-19-01423-t004:** Descriptive statistics for participant group and bottle type.

Participant Group	Bottle Type	N	Mean	Std. Deviation
Younger adults	A	34	3.59	1.05
	B	34	3.82	1.06
	C	34	2.62	1.02
	D	34	3.06	0.92
	E	34	3.00	0.89
Older adults	A	38	3.79	0.78
	B	38	4.08	0.82
	C	38	3.16	1.22
	D	38	3.61	0.68
	E	38	3.35	0.83
Total	A	72	3.69	0.91
	B	72	3.96	0.94
	C	72	2.90	1.15
	D	72	3.35	0.84
	E	72	3.19	0.87

**Table 5 ijerph-19-01423-t005:** Results of the two-way ANOVA for age group and bottle type.

Predictor	SS	*df*	MS	F	*p*
Ages	12.93	1	12.93	14.8	0
Bottle types	50.67	4	12.67	14.51	0
					A > C, E
					B > C, D, E
Ages Ⅹ Bottle types	1.83	4	0.46	0.52	0.72
Error	307.4	352	0.87		
Corrected total	4599	362			

SS: Sum of squares, MS: Mean square, *df*: Degree of freedom, Age X Bottle type: effect of the interaction between age and bottle type.

**Table 6 ijerph-19-01423-t006:** Results of the regression analysis and * indicated the significant levels (* < 0.05, ** < 0.001).

Group	Predictor		Unstandardised Coefficient		Standardised Coefficient	t	*p*	Tolerance	VIF
			B	SE	β				
Younger adults	Before-opening	Visual attractiveness	0.09	0.05	0.12	1.71	0.09	0.60	1.68
		Expected easiness	0.12	0.11	0.08	1.11	0.27	0.65	1.54
	After-opening	Satisfaction with bottle thickness	0.05	0.08	0.04	0.57	0.57	0.55	1.81
		Satisfaction with cap height	0.10	0.09	0.10	1.17	0.24	0.44	2.30
		Satisfaction with cap ridges	0.06	0.08	0.06	0.73	0.47	0.50	2.02
		Satisfaction with hand strength required	0.54	0.07	0.58	8.13	0.00 **	0.56	1.67
		Satisfaction with feeling of spilling out	0.05	0.06	0.58	0.82	0.41	0.62	1.61
Older adults	Before-opening	Visual attractiveness	−0.00	0.05	−0.01	−0.09	0.93	0.76	1.31
		Expected easiness	−0.05	0.10	−0.04	−0.51	0.61	0.70	1.43
	After-opening	Satisfaction with bottle thickness	0.18	0.07	0.21	2.76	0.01 *	0.64	1.57
		Satisfaction with cap height	0.10	0.11	0.11	0.92	0.36	0.27	3.69
		Satisfaction with cap ridges	0.13	0.08	0.12	1.66	0.10	0.68	1.47
		Satisfaction with hand strength required	0.20	0.07	0.22	2.81	0.01 *	0.60	1.68
		Satisfaction with feeling of spilling out	0.02	0.07	0.02	0.32	0.75	0.76	1.31

Younger adults: *R*^2^(adj. *R*^2^) = 0.50 (0.48), *F* 20.47, *p* = 0.00, Older adults: *R*^2^(adj. *R*^2^) = 0.33 (0.30), *F* 11.10, *p* = 0.00.

## Data Availability

The data that support the findings of this study are available on request from the corresponding author.

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
