# Peer review of "Designing for Green and Grey: Insights from Single-Use Plastic Water Bottles"

_ijerph, 2022, doi:10.3390/ijerph19031423_

Round 1
Reviewer 1 Report
The article is well written and scientific sound. Although, some minor corrections required:
- Fig 2, must be replaced by using bar graph with standard error.
- Please clarify Table 6, all p value higher than 5, means that all parameters are not significant
- Fig 1, give caption for each figure
- Fig 3, give caption with a, b and c for each figure
Author Response
1] Fig 2, must be replaced by using bar graph with standard error.
>> It was replaced with a bar chart with standard error.
2] Please clarify Table 6, all p value higher than 5, means that all parameters are not significant
>>Answer: The table shows that some of the factors, saying four factors, are significant. An important point is a difference of effective factors between the age groups. The significant levels are indicated in the Table. in the text, it is mentioned that
"For the younger adults, the important predictor was hand strength (β = .58, t = 8.13, p = .00), while for the older adults, significant predictors were bottle thickness (β = .21, t = 2.76, p = .01) and hand strength (β = .22, t = 2.81, p = .01). Table 6 provides the detailed results of the regression analysis”.
3] Fig 1, give caption for each figure
>> Answer: Captions have been provided.
4] Fig 3, give caption with a, b and c for each figure
>>Answer: Captions have been given
Reviewer 2 Report
The work by Kim and Lee reports on environmental and social issues related to single use plastic bottles. This is quite a relevant topic as environmental issues are getting more and more relevant while societies in OECD countries gets older. Thus, a compromise should be reached. In this manuscript these topics are discussed and a statistical research is presented. The work is well done and well written. Discussion is supported in their own results and in previous works. Therefore, I recommend its publication, but I have some comments and suggestions for authors as follows:
Would you expect any effect on the type of plastic? Your study seems to be performed on PET bottles, which are the most common ones nowadays, but in the future probably biodegradable polymers will become an option.
Did you observed any influence for right-handed vs lest-handed individuals in your study?
Have you considered bottle openers that are quite common in some societies for elderly people?
Author Response
1] Would you expect any effect on the type of plastic? Your study seems to be performed on PET bottles, which are the most common ones nowadays, but in the future probably biodegradable polymers will become an option.
>> Answer: Yes, I agree with you. When we have the bottles of biodegradable polymers decreasing the plastic bottles’ environmental problems, the weak openability issue would be more stood out. Thus, this study can be a good starting point for better ecological matters. (It is included in the Discussion. Please, see section 4.2, the line numbers #362-366).
2] Did you observed any influence for right-handed vs lest(left)-handed individuals in your study?
>> Answer: Considering the situation using both hands to open bottles in ordinary settings, using right-handed vs. left hand can be like a mirror image. Hence, the resultant difference seems not so significant. However, we may have a chance to think about it in a future study. Thank you for your suggestion.
3] Have you considered bottle openers that are quite common in some societies for elderly people?
>>Answer: This paper mentions a little as related to the issue (line numbers #112-119).
“Even though older people and even some young and non-disabled adults struggle to open the bottles, our society has neglected their difficulties. Industries have avoided addressing the issue by offering various jar openers, from small rubber discs to expensive gadgets. Nonetheless, there are no perfect tools to satisfy people who need help with grip, two strong hands, leverage strength, or all three at once. Each tool or device can help only some people by accommodating a particular type of difficulty. The countless tools themselves serve as evidence of the magnitude of this problem”.
Thank you for your point.
Reviewer 3 Report
Dear Authors,
thank you for a rich and interesting overview of the problem.
In order to improve readership and future citation of your excellent paper, I suggest the following:
1) Cut the text.
For instance, sub-chapters 1.1 and 1.2. do not have a direct connection to the problem of bottles opening and an easy-to-use system for elderly. In fact, these two sub-chapters can be a part of another paper - a review of an aging population. Please, remove them (cut about 2 pages of the text) but add the most significant parts of these chapters to the revised Introduction.
2) Add a comment about your recruited participants, why did you select these groups?
My general comments: you don't have participants in the group of 51-60. Why? This seems to be an important group.
Thank you and good luck.
Author Response
thank you for a rich and interesting overview of the problem.
In order to improve readership and future citation of your excellent paper, I suggest the following:
1) Cut the text.
For instance, sub-chapters 1.1 and 1.2. do not have a direct connection to the problem of bottles opening and an easy-to-use system for elderly. In fact, these two sub-chapters can be a part of another paper - a review of an aging population. Please, remove them (cut about 2 pages of the text) but add the most significant parts of these chapters to the revised Introduction.
>>Answer: I understand the reviewer's point, but at the same time, I think it is crucial to show a situation in Korea, greying but not prepared, entering the second stage that asks for the private sector's participation to build enabling environment for the elderly.
Thus, taking the reviewer's opinion, the two chapters were merged and made into one shorter chapter (Please, see section 1.1).
2) Add a comment about your recruited participants, why did you select these groups?
My general comments: you don't have participants in the group of 51-60. Why? This seems to be an important group.
>>Answer: The subjects were gathered through convenience and snowball sampling, and thus the absence of the age group happened unintentionally. There were no additional efforts to get people in the age group because this absence does not significantly impact research results. Related to this issue, some explanation has been added as follows (please, see line numbers #218-224);
"The initial subjects of the younger group were contacted through the author's network and then asked to recruit others through their networks. As a result, unintentionally, there was none of the age group of 51-60. There were no additional efforts to gather the age group members considering the absence of this age group would not cause any significant effects in terms of muscle loss with aging. Muscle mass begins to decrease around 50, but it accelerates after 60. The rate of physiological changes from aging in the elderly over 65 is more than 25%, and over 80 is about 50% [48]".